# Optimal Training of Fair Predictive Models

**Razieh Nabi**        RAZIEH.NABI@EMORY.EDU
*Department of Biostatistics and Bioinformatics, Emory University*

**Daniel Malinsky**        D.MALINSKY@COLUMBIA.EDU
*Department of Biostatistics, Columbia University*

**Ilya Shpitser**        ILYAS@CS.JHU.EDU
*Department of Computer Science, Johns Hopkins University*

**Editors:** Bernhard Schölkopf, Caroline Uhler and Kun Zhang

## Abstract

Recently there has been sustained interest in modifying prediction algorithms to satisfy fairness constraints. These constraints are typically complex nonlinear functionals of the observed data distribution. Focusing on the path-specific causal constraints proposed by Nabi and Shpitser (2018), we introduce new theoretical results and optimization techniques to make model training easier and more accurate. Specifically, we show how to reparameterize the observed data likelihood such that fairness constraints correspond directly to parameters that appear in the likelihood, transforming a complex constrained optimization objective into a simple optimization problem with box constraints. We also exploit methods from empirical likelihood theory in statistics to improve predictive performance by constraining baseline covariates, without requiring parametric models. We combine the merits of both proposals to optimize a hybrid reparameterized likelihood. The techniques presented here should be applicable more broadly to fair prediction proposals that impose constraints on predictive models.

**Keywords:** Algorithmic fairness, causal inference, counterfactual path-specific effects.

## 1. Introduction

Predictive models trained on imperfect data are increasingly being used in socially-impactful settings. Predictions (such as risk scores) have been used to inform high-stakes decisions in criminal justice (Perry et al., 2013), healthcare (Kappen et al., 2018), and finance (Khandani et al., 2010). While automation may bring many potential benefits – such as speed and accuracy – it is also fraught with risks. Predictive models introduce two dangers in particular: the illusion of objectivity and violation of fairness norms. Predictive models may appear to be "neutral," since humans are less involved and because they are products of a seemingly impartial optimization process. However, predictive models are trained on data that reflects the structural inequities, historical disparities, and other imperfections of our society. Often data includes sensitive attributes (e.g., race, gender, age, disability status), or proxies for such attributes. A particular worry in the context of data-driven decision-making is "perpetuating injustice," which occurs when unfair dependence between sensitive features and outcomes is maintained, introduced, or reinforced by automated tools.

We study how to construct fair predictive models by correcting for the unfair causal dependence of predicted outcomes on sensitive features. We work with the proposed fairness criteria in Nabi and Shpitser (2018), where the fair prediction requires imposing hard constraints on the predictive model in the form of restricting certain causal path-specific effects. Impermissible pathways are user-specified and context-specific, hence the framework requires input from policymakers, legal

experts, or the general public. Some alternative but also causally-motivated constrained prediction methods are proposed in Kusner et al. (2017a); Zhang and Bareinboim (2018); Chiappa (2019). For a survey and discussion of distinct fairness criteria (both causal and associative) see Mitchell et al. (2018).

We advance the state of the art in two ways. First, we give a novel reparameterization of the observed data likelihood in which unfair path-specific effects appear directly as parameters. This allows us to greatly simplify the constrained optimization problem, which has previously required complex or inefficient algorithms. Second, we demonstrate how tools from the empirical likelihood literature (Owen, 2001) can be readily adapted to construct hybrid (semi-parametric) observed data likelihoods that satisfy given fairness criteria. With this approach, the entire likelihood is constrained, rather than only part of the likelihood as in past proposals. As a result, we use the data more efficiently and achieve better performance. Finally, we show how both innovations may be combined into a single procedure.

As a guiding example we consider computer-assisted hiring, in which predictive models are used to infer job success from features found in applicant data. In this setting, we assume models have access to historical data on job success of some applicants, quantified by a numerical score, as well as their resume information including demographics. In addition, we are interested in predicting the job success score of new individuals for whom only resume and demographic information is available. This may be considered a variant of semi-supervised learning or prediction with missing labels on a subset of the population. We aim to estimate scores quantifying job success subject to path-specific fairness constraints that ensure, for instance, that perceived race has no direct influence on the component of the job success score pertaining to employee evaluation by their supervisor. In order to describe the various components of this proposal, we must review some background on causal inference, path-specific effects, and constrained prediction.

## 2. Causal Inference and a Causal Approach to Fairness

Causal inference is concerned with quantities which describe the consequences of interventions. Causal models are often represented graphically, e.g. by directed acyclic graphs (DAGs). We will use capital letters ($V$) to denote sets of random variables as well as corresponding vertices in graphs and lowercase letters ($v$) to denote values or assignments to those random variables. A DAG consists of a set of vertices $V$ connected by directed edges ($V_i \rightarrow V_j$ for some $\{V_i, V_j\} \subseteq V$) such that there are no directed cycles. The set $\mathrm{pa}_{\mathcal{G}}(V_i) \equiv \{V_j \in V \mid V_j \rightarrow V_i\}$ denotes the parents of $V_i$ in DAG $\mathcal{G}$. $\mathfrak{X}_A$ denotes the statespace of $A \subseteq V$.

A causal model of a DAG $\mathcal{G}$ is a set of distributions defined on potential outcomes (a.k.a. counterfactuals). For example, we consider distributions $p(V(a))$ subject to some restrictions, where $V(a)$ represents the value of $V$ had all variables in $\mathrm{pa}_{\mathcal{G}}(V)$ been set, possibly contrary to fact, to value $a$. In this paper, we assume Pearl's *structural causal model* (Pearl, 2009) for a DAG $\mathcal{G}$ which stipulates that the sets of potential outcome variables $\left\{\{V_i(a_i) \mid a_i \in \mathfrak{X}_{\mathrm{pa}_{\mathcal{G}}(V_i)}\} \mid V_i \in V\right\}$ are mutually independent. All other counterfactuals may be defined using *recursive substitution*:

$$V_i(a) \equiv V_i(a_{\mathrm{pa}_{\mathcal{G}}(V_i) \cap A}, \{V_j(a) : V_j \in \mathrm{pa}_{\mathcal{G}}(V_i) \setminus A\}), \quad \text{for any } A \subseteq V \setminus \{V_i\}.$$

where $\{V_j(a) : V_j \in \mathrm{pa}_{\mathcal{G}}(V_i) \setminus A\}$ is taken to mean the (recursively defined) set of counterfactuals associated with variables in $\mathrm{pa}_{\mathcal{G}}(V_i) \setminus A$, had $A$ been set to $a$. Equivalently, Pearl's model may be described by a system of nonparametric structural equations with independent errors.

A causal parameter is said to be *identified* in a causal model if it is a function of the observed data distribution $p(V)$. In the structural causal model of a DAG $\mathcal{G}$ (as well as some weaker causal models), all interventional distributions $p(V(a))$, for any $A \subseteq V$, are identified by the *g-formula*: $p(V(a)) = \prod_{V_i \in V \setminus A} p(V_i | \mathrm{pa}_{\mathcal{G}}(V_i))\big|_{A=a}$. For example, consider the DAG in Fig. 1(a). $Y(a)$ is defined to be $Y(a, M(a, X), X)$ by recursive substitution and its distribution is identified as $\sum_{X,M} p(Y|a, M, X) \times p(M|a, X) \times p(X)$. The mean difference between $Y(a)$ and $Y(a')$ for some treatment value $a$ of interest and reference value $a'$ is $\mathbb{E}[Y(a)] - \mathbb{E}[Y(a')]$ and quantifies the *average causal effect* of treatment $A$ on the outcome $Y$.

## 2.1. Mediation Analysis and Path-Specific Effects

An important goal in causal inference is to understand the mechanisms by which some treatment $A$ influences some outcome $Y$. A common framework for studying mechanisms is *mediation analysis* which seeks to decompose the effect of $A$ on $Y$ into the *direct effect* and the *indirect effect* mediated by a third variable, or more generally into components associated with particular causal pathways. As an example, the direct effect of $A$ on $Y$ in Fig. 1(a) corresponds to the effect along the edge $A \rightarrow Y$ and the indirect effect corresponds to the effect along the path $A \rightarrow M \rightarrow Y$, mediated by $M$.

In the potential outcome notation, the direct and indirect effects can be defined using nested counterfactuals such as $Y(a, M(a'))$ for $a, a' \in \mathfrak{X}_A$, which denotes the value of $Y$ when $A$ is set to $a$ while $M$ is set to whatever value it would have attained had $A$ been set to $a'$. The *natural direct effect* (NDE) (on the expectation difference scale) is defined as $\mathbb{E}[Y(a, M(a'))] - \mathbb{E}[Y(a')]$ and the *natural indirect effect* (NID) is defined as $\mathbb{E}[Y(a)] - \mathbb{E}[Y(a, M(a'))]$. Under certain identification assumptions discussed by Pearl (2001), the distribution of $Y(a, M(a'))$ (and thereby direct and indirect effects) can be nonparametrically identified from observed data by the following formula:

$$p(Y(a, M(a'))) = \sum_{X,M} p(Y \mid a, X, M) \, p(M \mid a', X) \, p(X).$$

More generally, when there are multiple *proper* pathways from $A$ to $Y$ (a proper causal path only intersects $A$ at the source node) one may define various *path-specific effects* (PSEs). The effect along a specific path will be obtained by comparing two potential outcomes, one where for the selected paths all nodes behave as if $A = a$, and along all other paths nodes behave as if $A = a'$.

PSEs are defined by means of nested, path-specific potential outcomes. Fix a set of treatment variables $A$, and a subset of *proper causal paths* $\pi$ from any element in $A$. Next, pick a pair of value sets $a$ and $a'$ for elements in $A$. For any $V_i \in V$, define the potential outcome $V_i(\pi, a, a')$ by setting $A$ to $a$ for the purposes of paths in $\pi$, and to $a'$ for the purposes of proper causal paths from $A$ to $Y$ not in $\pi$. Formally, the definition is as follows, for any $V_i \in V$, $V_i(\pi, a, a') \equiv a$ if $V_i \in A$, otherwise

$$V_i(\pi, a, a') \equiv V_i\Big(\big\{V_j(\pi, a, a') \mid V_j \in \mathrm{pa}_{\mathcal{G}}^{\pi}(V_i)\big\}, \big\{V_j(a') \mid V_j \in \mathrm{pa}_{\mathcal{G}}^{\overline{\pi}}(V_i)\big\}\Big), \tag{1}$$

where $V_j(a') \equiv a'$ if $V_j \in A$ and given by recursive substitution otherwise, $\mathrm{pa}_{\mathcal{G}}^{\pi}(V_i)$ is the set of parents of $V_i$ along an edge which is a part of a path in $\pi$, and $\mathrm{pa}_{\mathcal{G}}^{\overline{\pi}}(V_i)$ is the set of all other parents of $V_i$.

A counterfactual $V_i(\pi, a, a')$ is said to be *edge inconsistent* if counterfactuals of the form $V_j(a_k, \ldots)$ and $V_j(a'_k, \ldots)$ occur in $V_i(\pi, a, a')$, otherwise it is said to be *edge consistent*. It is known that a joint distribution $p(V(\pi, a, a'))$ containing an edge-inconsistent counterfactual $V_i(\pi, a, a')$

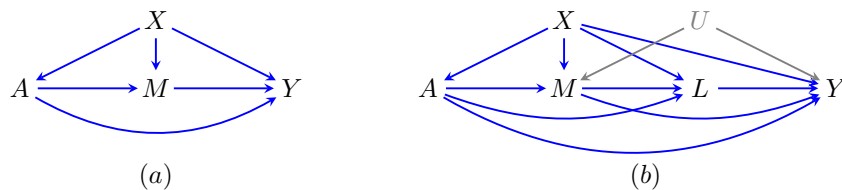

Figure 1: (a) A causal DAG, with treatment $A$, outcome $Y$, baseline variables $X$, and a mediator $M$. (b) A causal graph with two mediators $M$ and $L$ and unmeasured confounders $U$.

is not identified in the structural causal model (nor weaker causal models) with a corresponding graphical criterion on $\pi$ and $\mathcal{G}(V)$ called the 'recanting witness' (Shpitser, 2013). Under some assumptions, PSEs are nonparametrically identified by means of the *edge g-formula* described in Shpitser and Tchetgen Tchetgen (2016) and reproduced in our Appendix A.

As an example, consider the DAG in Fig. 1(b). The PSE of $A$ on $Y$ along the paths $\pi = \{A \rightarrow Y, A \rightarrow L \rightarrow Y\}$ is encoded by a counterfactual contrast of the form $Y(\pi, a, a') = Y(a, M(a'), L(a, M(a')))$. The corresponding counterfactual density is identified by a special case of the edge g-formula as follows: (for more details on PSEs, see Shpitser (2013))

$$p(Y(a, M(a'), L(a, M(a')))) = \sum_{X,M,L} p(Y \mid a, X, M) \times p(L \mid a, M, X) \times p(M \mid a', X) \times p(X).$$

### 2.2. Algorithmic Fairness via Constraining Path-Specific Effects

There has been a growing interest in the issue of fairness in machine learning (Pedreshi et al., 2008; Feldman et al., 2015; Hardt et al., 2016; Kamiran et al., 2013; Corbett-Davies et al., 2017; Jabbari et al., 2017; Kusner et al., 2017b; Zhang and Bareinboim, 2018; Zhang et al., 2017; Gillis et al., 2021). In this paper, we adopt the causal notion of fairness described in Nabi and Shpitser (2018) and Nabi et al. (2019), where unfairness corresponds to the presence of undesirable or impermissble path-specific effects of sensitive attributes on outcomes – a view which generalizes an example discussed in Pearl (2009). We provide a brief summary of their perspective on fairness in the following without defending it for lack of space; see Nabi and Shpitser (2018) for more details.

Consider an observed data distribution $p(Y, Z)$ induced by a causal model, where $Y$ is an outcome and $Z = \{X, A, M\}$ includes all baseline factors $X$, sensitive features $A$, and post-treatment pre-outcome mediators $M$. Context and background ethical considerations pick out some path-specific effect of the sensitive feature $A$ on the outcome $Y$ as unfair. We assume this effect is identified as some function of the observed distribution: $g(p(Y, Z))$. Fix upper and lower bounds $\epsilon_l, \epsilon_u$ for the PSE, representing a tolerable range. The most relevant bounds in practice are $\epsilon_l = \epsilon_u = 0$ or approximately zero. Nabi and Shpitser propose to transform the inference problem on $p(Y, Z)$, the "unfair world," into an inference problem on another distribution $p^*(Y, Z)$, called the "fair world," which is close in the sense of minimal KL-divergence to $p(Y, Z)$ while also having the property that the PSE lies within $(\epsilon_l, \epsilon_u)$.

Given a dataset $\mathcal{D} = \{(Y_i, Z_i), i = 1, \ldots, n\}$ drawn from $p(Y, Z)$, a likelihood function $\mathcal{L}(\mathcal{D}; \alpha)$ parameterized by $\alpha$, an estimator $\widehat{g}(\mathcal{D})$ of the unfair PSE, and bounds $\epsilon_l, \epsilon_u$, Nabi and Shpitser (2018)

suggest to approximate $p^*(Y, Z)$ by solving the following constrained maximum likelihood problem:

$$\widehat{\alpha} = \arg\max_{\alpha} \ \mathcal{L}_{Y,Z}(\mathcal{D}; \alpha) \quad \text{subject to} \ \ \epsilon_l \leq \widehat{g}(\mathcal{D}; \alpha) \leq \epsilon_u. \tag{2}$$

(Here we write the estimated PSE as $\widehat{g}(\mathcal{D}; \alpha)$ to make explicit that this constraint depends on parameters which parameterize the likelihood function.) Having approximated the fair world $p^*(Y, Z; \widehat{\alpha})$ in this way, Nabi and Shpitser (2018) point out a key difficulty for using these estimated parameters to predict outcomes for new instances (e.g., new job applicants). A new set of observations $Z$ is not sampled from the "fair world" $p^*(Z)$ but from "unfair world" $p(Z)$. Nabi and Shpitser (2018) propose to map new instances from $p$ to $p^*$ and use the result for predicting $Y$ with constrained model parameters $\widehat{\alpha}$. They assume $Z$ can be partitioned into $Z_1$ and $Z_2$ such that $p^*(Y, Z) = p^*(Y, Z_1 | Z_2) p(Z_2)$. In other words, variables in $Z_2$ are shared between $p$ and $p^*$: $p^*(Z_2) = p(Z_2)$ but $p^*(Z_1 | Z_2) \neq p(Z_1 | Z_2)$. $Z_1$ typically corresponds to variables that appear in the estimator $\widehat{g}(\mathcal{D})$. There is no obvious principled way of knowing exactly what values of $Z_1$ the "fair version" of the new instance would attain. Consequently, all such possible values are averaged out, weighted appropriately by how likely they are according to the estimated $p^*$. This entails predicting $Y$ as the expected value $\mathbb{E}^*[Y | Z_2]$, with respect to the distribution $\sum_{Z_1} p^*(Y, Z_1 | Z_2)$.

Next, we explain some limitations of the inference procedure described here and present our main contributions to address these limitations.

## 3. Fair Predictive Models in a Batch Setting

Prediction problems in machine learning are typically tackled from the perspective of nonparametric risk minimization and the "train-and-test" framework. Here, we instead take the perspective of maximum likelihood and missing data, i.e., we treat unknown outcomes as missing values which we hope to impute in a way that is consistent with our specified likelihood for the entire data set. Our motivation for doing so is the nature of our constrained prediction problem. Specifically, our causal constraints contain "nuisance" components (conditional expectations and conditional distributions derived from the observed data distribution) which must be modeled correctly to ensure the causal effects are reliably estimated. Specifically, we choose to estimate these nuisance components (semi-)parametrically because we desire certain frequentist properties, namely fast rates of convergence, for estimating the relevant PSEs. In the subsequent prediction step, we should predict in a way that is consistent with what has already been modeled, or else we fail to exploit all the information we have already committed to in the constraint estimation step. We choose the maximum likelihood framework as the most natural and simplest approach to accomplish this. Alternative methods for coherently combining nuisance estimation with nonparametric risk minimization are left to future work.

Unlike Nabi and Shpitser (2018), we consider a batch prediction setting – this allows us to avoid the inefficient averaging described in the previous section. In our case, historical data (of sample size $n_1$) consists of observations on $\{X, A, M, Y\}$ and new instances (of size $n_2$) comprise a set of observations with just $\{X, A, M\}$. The outcome labels for new instances are missing data which we aim to predict, subject to fairness constraints. Instead of training our constrained model on historical data alone, we train on the combination of historical data and new instances. This seems complicated since the observed data likelihood for the combined data set includes some complete rows and some partially incomplete rows. However, we can borrow ideas from the literature on missing data to accomplish this task. Specifically, we can impute missing outcomes

("labels") using appropriate functions of observed data. In this paper we assume the labels are *missing at random* (MAR), as is typical in the semi-supervised learning setting (Little and Rubin, 2002; Lafferty and Wasserman, 2008). Specifically, we assume that the instances with missing labels are sampled from the same distribution that generated the complete historical data (with observed labels) – this satisfies MAR, since whether a label is missing or not for a particular instance is not informative. Let the random variable $R$ denote the missingness status of the outcome variable $Y$ for each instance. That is, $R = 1$ for all rows in the historical data (since $Y$ is observed) and $R = 0$ for all rows in the new instances. Then, assuming MAR, the observed data likelihood is $\prod_{i=1}^{n=n_1+n_2} p(X_i, A_i, M_i) \, p(R_i|X_i, A_i, M_i) \, p(Y_i|X_i, A_i, M_i)^{R_i}$. Our approach may be extended to any identifiable *missing not at random* (MNAR) model by appropriately modifying this observed data likelihood. See Bhattacharya et al. (2019); Malinsky et al. (2021); Nabi et al. (2020) for recent developments on identified MNAR models.

The likelihood function describes the probability of the entire data set, though it only uses $Y$ values from historical data. We can then maximize the likelihood subject to the specified path-specific constraints, and associate predicted values $\widehat{Y}_{new}$ to the new instances. Note that the setting where new instances arrive sequentially one-at-a-time is a special case of this general setup, which would require retraining on the full combined data after the arrival of each instance. Though this is computationally more intensive than the proposal in Nabi and Shpitser (2018) (where they only train once), it will deliver significantly more accurate predictions because it uses all available information. We will elaborate on this point in Section 4.

The approach to fair prediction outlined in Nabi and Shpitser (2018) suffers from two problems: one general and one specific to our setting here. First, their approach requires solving a computationally challenging constrained optimization problem. The constraints on path-specific effects involve nonlinear and complicated functionals of the observed data distribution. This makes the proposed constrained optimization a daunting task that relies on complex optimization software (or computationally expensive methods such as rejection sampling), which do not always find high quality local optima. Second, Nabi and Shpitser (2018) propose to constrain only part of the likelihood. Specifically they do not constrain the density $p(X)$ over the baseline features (since this is high-dimensional and thus inplausible to model accurately in their parametric approach). The baseline density is instead estimated by placing $1/n$ mass at every observed data point. This is sub-optimal in the specific setting we consider, where we do not need to average over constrained variables. Constraining a larger part of the joint distribution should lead to a fair world distribution KL-closer to the observed distribution, which leads to better predictive performance as long as the likelihood is correctly specified. This intuition is formalized in the following result.

**Theorem 1** *Let $p(Z)$ denote the observed data distribution and let $Z_1, Z_2 \subseteq Z$. Let $p_1^*(Z)$ and $p_2^*(Z)$ denote two constrained distributions that are obtained by constraining the distribution over variables that are* not *in $Z_1$ and $Z_2$, respectively. That is $p_1^*$ constrains the variables in $Z \setminus Z_1$ and $p_2^*$ constrains the variables in $Z \setminus Z_2$, so $p_1^*(Z_1) = p(Z_1)$ and $p_2^*(Z_2) = p(Z_2)$. More formally,*

$$p_1^*(Z) = \underset{q(Z)}{\arg\min} \ D_{KL}(p \,\|\, q), \ \text{s.t. } \epsilon_l \leq g(q(Z)) \leq \epsilon_u, \text{ and } q(Z_1) = p(Z_1),$$

$$p_2^*(Z) = \underset{q(Z)}{\arg\min} \ D_{KL}(p \,\|\, q), \ \text{s.t. } \epsilon_l \leq g(q(Z)) \leq \epsilon_u, \text{ and } q(Z_2) = p(Z_2).$$

*If $Z_2 \subseteq Z_1 \subseteq Z$, then $D_{KL}(p \,\|\, p_2^*) \leq D_{KL}(p \,\|\, p_1^*)$.*

The conditions in Theorem 1 specify that a larger part of $p_2^*(Z)$ is constrained compared to $p_1^*(Z)$ (since $Z_2 \subseteq Z_1 \subseteq Z$ implies that $\{Z \setminus Z_1\} \subseteq \{Z \setminus Z_2\} \subseteq Z$). Theorem 1 then states that $p_2^*(Z)$ is at least as close to $p(Z)$ as $p_1^*(Z)$. This should match intuition: if a larger part of the joint is being constrained, there are more "degrees of freedom" available to satisfy the constraint, and so the constrained distribution may lie "closer" to the unconstrained distribution.

To address the first aforementioned difficulty (that constraints are complex nonlinear functions of the joint), we provide a novel reparameterization of the observed data likelihood such that the causal parameter corresponding to the unfair PSE appears directly in the likelihood. This approach generalizes previous work on reparameterizations implied by structural nested models (Robins, 2000; Tchetgen Tchetgen and Shpitser, 2014) to apply to a wide class of PSEs. With such a reparameterization, the MLE with a PSE constraint simply corresponds to maximizing the likelihood in a submodel where a certain likelihood parameter is set to $0$. Optimization can then be carried out with standard software.

To address the second difficulty (that constraining only part of the likelihood is suboptimal), we propose an approach to constraining the density $p(X)$. An alternative to fully parametric modeling is to consider nonparametric representations of $p(X)$. It is well known that the nonparametric maximum likelihood estimate of any $p(X)$ from a set of i.i.d draws is the empirical distribution, placing mass $1/n$ at every observed point. Empirical likelihood methods have been developed for settings where the nonparametric and parametric (hybrid) likelihood must be maximized subject to moment constraints (Owen, 2001). We describe below how these methods may be adapted to our setting, taking advantage of the fact that constraints on the PSEs we consider correspond to moment constraints. Finally, we show how both the reparameterization method and the empirical likelihood method can be combined to yield a constrained optimization method that maximizes a semi-parametric (hybrid reparameterized) likelihood.

## 4. Efficient Approximation of Fair Worlds

### 4.1. Fairness Constraints Via Reparameterized Likelihoods

In this section, we describe how to reparameterize the observed data likelihood in terms of causal parameters that correspond to path-specific effects. The result presented in the following theorem greatly simplifies the constrained optimization problem (2) in settings where the PSE includes the direct influence of $A$ on $Y$. This is due to the fact that the constrained parameter, corresponding to the PSE of interest, now appears as a single coefficient in the outcome regression model. For simplicity, we describe the reparameterization approach without reference to missing data – the extension to missing at random models we use in our analysis is straightforward.

**Theorem 2** *Assume the observed data distribution $p(Y, Z)$ is induced by a causal model where $Z = \{X, A, M\}$ includes pre-treatment measures $X$, binary treatment $A$, and post-treatment pre-outcome mediators $M$. Let $p(Y(\pi, a, a'))$ denote the potential outcome distribution that corresponds to the effect of $A$ on $Y$ along proper causal paths in $\pi$, where $\pi$ includes the direct edge $A \to Y$, and let $p(Y_0(\pi, a, a'))$ denote the identifying functional for $p(Y(\pi, a, a'))$ obtained from the edge g-formula, where the term $p(Y|Z)$ is evaluated at $\{Z \setminus A\} = 0$. Then $\mathbb{E}[Y|Z]$ can be written as:*

$$\mathbb{E}[Y|Z] = f(Z) - \big(\mathbb{E}[Y(\pi, a, a')] - \mathbb{E}[Y_0(\pi, a, a')]\big) + \phi(A),$$

where $f(Z) := \mathbb{E}[Y|Z] - \mathbb{E}[Y|A, \{Z \setminus A\} = 0]$ *and* $\phi(A) = w_0 + w_a A$. *Furthermore,* $w_a$ *corresponds to $\pi$-specific effect of $A$ on $Y$.*

To illustrate the above reparameterization, consider the graph in Fig. 1(b), discussed in Nabi and Shpitser (2018); Chiappa (2019). Assume the direct path and the paths through $M$ of $A$ on $Y$ are the impermissible pathways. The corresponding PSE is encoded by a counterfactual contrast with respect to $Y(a, M(a), L(a', M(a)))$. The reparameterization in Theorem 2 amounts to:

$$\mathbb{E}[Y \mid Z] = f(Z) - \sum_{Z \setminus A} \left\{ f(Z) \times p(L \mid M, X, A = 0) \times p(M \mid X, A = 1) \times p(X) \right\} + w_0 + w_a A, \quad (3)$$

where $w_a$ represents the PSE of interest and $f(Z) := \mathbb{E}[Y \mid Z] - \mathbb{E}[Y \mid A, X = M = L = 0]$; see the Appendix A for a detailed derivation.

Under linearity assumptions, the PSE of interest in Fig. 1(b) has a simple form. Assume the data generating process in Fig. 1(b) is the same as the one given in display (2) of Chiappa (2019), i.e., a system of linear equations with $\theta_k^j$ denoting the linear coefficent on variable $k$ in the structural equation for variable $j$. Then by simple path analysis, PSE $= \theta_a^y + \theta_m^y \theta_a^m + \theta_l^y \theta_m^l \theta_a^m$. In this case, our reparameterization takes the following form:

$$\mathbb{E}[Y \mid X, A, M, L] = \underbrace{\left( \theta_x^y X + \theta_m^y M + \theta_l^y L \right)}_{f(Z)} - \underbrace{\left( \left( \theta_0^m \theta_m^y + (\theta_0^l + \theta_m^l \theta_0^m)\theta_l^y \right) + \left( \theta_m^y \theta_a^m + \theta_l^y \theta_m^l \theta_a^m \right) A \right)}_{\sum_{Z \setminus A} \left\{ f(Z) \times p(L|M,X,A=0) \times p(M|X,A=1) \times p(X) \right\}} +$$

$$\underbrace{\left( \theta_0^y + \left( \theta_0^m \theta_m^y + (\theta_0^l + \theta_m^l \theta_0^m)\theta_l^y \right) \right)}_{w_0} + \underbrace{\left( \theta_a^y + \theta_m^y \theta_a^m + \theta_l^y \theta_m^l \theta_a^m \right) A}_{w_a \equiv \text{PSE}} \Big).$$

In order to move away from the linear setting and exploit more flexible techniques, Chiappa (2019) posits some assumptions on the latent variables. However, such assumptions are often hard to verify in practice. In contrast, our result in identifying the PSE is entirely nonparametric and does not rely on any assumptions beyond what is encoded in the causal DAG.

Given Theorem 2, the constrained optimization problem in eq. (2) significantly simplifies to the following optimization problem:

$$\widehat{\alpha} = \arg\max_{\alpha} \ \mathcal{L}_{Y,Z}(\mathcal{D}; \alpha) \quad \text{subject to} \ \epsilon_l \le w_a \le \epsilon_u, \quad (4)$$

where $\alpha$ contains $w_a$ and the nonlinear constraint has been replaced by a box-constraint on the parameter $w_a$. In the prediction setting, i.e., finding optimal parameters for $\mathbb{E}[Y|Z; \alpha_y]$, this amounts to an unconstrained maximum likelihood problem with outcome regression taking the specific form where $w_a$ is set to zero. For instance, the regression in eq. (3) becomes $\mathbb{E}[Y|Z; \alpha_y] = f(Z; \alpha_f) - \sum_{X,M,L} \left\{ f(Z; \alpha_f) \times p(L|M, X, A = 0; \alpha_m) \times p(M|X, A = 1; \alpha_m) \times p(X) \right\} + w_0$.

Likelihood reparameterization has been introduced for sequential ignorable models Robins (2000), but has not been studied in general for arbitrary path-specific effects. In addition to the immediate application in this paper, Theorem 2 solves a general open problem in generalizing structural nested models to longitudinal mediation analysis. A special case of this reparameterization, where PSE is simply the direct effect, is implicit in the work of Tchetgen Tchetgen and Shpitser

(2014). An advantage of this theorem in causal inference is developing flexible semiparametric estimators for arbitrary PSEs. With fairness being the primary focus of this paper, we do not investigate further the importance of this theorem in causal inference applications.

In the next section, we explain how $p(X)$ can be incorporated into the constrained optimization problem using empirical likelihood methods.

### 4.2. Fairness Constraints Via Hybrid Likelihoods

In light of Theorem 1, we are interested in constraining the nonparameteric form of $p(X)$. Following work in Owen (2001), we use hybrid/semi-parametric empirical likelihood methods to estimate $p(X)$ nonparametrically which is a novel idea in the fairness setting. First, according to Theorem 1, constraining $p(X)$ would bring our learned distribution closer to the observed (unfair) distribution, and hence results in improvement of model performance, as we demonstrate in our simulations. Second, $p(X)$ is often a high dimensional object that is difficult to estimate due to the curse of dimensionality. For simplicity of presentation, we focus on the DAG in Fig. 1(a), and the constraint represented by the NDE, although the methods we describe generalize without difficulty to arbitrary causal models and constraints represented by arbitrary PSEs.

Let $(X_i, A_i, M_i, Y_i), i = 1, \ldots, n$, be independent and identically distributed random vectors. We assume a semiparametric model on the joint distribution $p(Y, M, A, X)$ where $p(X)$ is left completely unspecified. If the unfair effect is the NDE, our constraint on the observed distribution is that the NDE should be zero. Let $p(Y|M, A, X), p(M|A, X), p(A|X)$ be parameterized by $\alpha = \{\alpha_y, \alpha_m, \alpha_a\}$. The direct effect can be identified by $\mathbb{E}_x[m(X; \alpha)]$, where

$$m(X; \alpha) = \sum_M \left\{ \mathbb{E}[Y|A = 1, M, X; \alpha_y] - \mathbb{E}[Y|A = 0, M, X; \alpha_y] \right\} \times p(M|A = 0, X; \alpha_m). \quad (5)$$

As is standard in empirical likelihood theory (we provide a general overview in Appendix B), we introduce "weight" parameters $p_i = p(X_i = x_i)$. The *profile empirical likelihood ratio* estimates $(\{\widehat{p}_i, \widehat{\alpha}\}^{opt})$ are then given by

$$\underset{p_i, \alpha}{\arg\max} \prod_{i=1}^n p_i \times p(Y_i \mid M_i, A_i, X_i; \alpha_y) \times p(M_i \mid A_i, X_i; \alpha_m) \times p(A_i \mid X_i; \alpha_a)$$

$$\text{such that} \quad \sum_{i=1}^n p_i = 1, \quad \sum_{i=1}^n p_i \times m(X_i; \alpha) = 0. \quad (6)$$

The above optimization problem involves a semi-parametric *hybrid* likelihood (Owen, 2001), that contains both nonparametric and parametric terms. In order to solve the above optimization problem (formulated on both $\alpha$ and $p_i$ parameters), we can apply the Lagrange multiplier method and solve its dual form (formulated on both $\alpha$ and the Lagrange multipliers); see Appendix B for more details. Empirical likelihood methods provide a natural extension to imposing constraints on arbitrary PSEs, since these can be written in the form of $\mathbb{E}_x[m(X; \alpha)]$ for some $m(\cdot)$.

If outcomes are missing at random, the NDE is identified by $\mathbb{E}_x[m_{\mathrm{mar}}(X; \alpha)]$, where

$$m_{\mathrm{mar}}(X; \alpha) \quad (7)$$
$$= \sum_M \left\{ \mathbb{E}[Y|A = 1, M, X, R = 1; \alpha_y] - \mathbb{E}[Y|A = 0, M, X, R = 1; \alpha_y] \right\} \times p(M|A = 0, X; \alpha_m).$$

---

**Algorithm 1** Hybrid Reparameterized Likelihood

---

**Input:** $\mathcal{D} = \{X_i, A_i, M_i, R_i, Y_i\}, i = 1, \ldots, n$ and specification of a PSE of the form $\mathbb{E}_X[m(X; \alpha)]$.
**Output:** $\widehat{\alpha}, \widehat{p}_i$ by solving

$$\underset{p_i, \alpha}{\arg\max} \sum_{i=1}^{n} \big( \log p_i + R_i \times \log p(Y_i \mid M_i, A_i, X_i; \alpha) + \log p(R_i, M_i, A_i \mid X_i; \alpha) \big)$$
$$\text{such that} \quad \sum_{i=1}^{n} p_i \times m_{\mathrm{mar}}(X_i, ; p_i, \alpha) = 0, \ \sum_{i=1}^{n} p_i = 1.$$

1: Pick starting values for $p_i^{(1)}$ and $\alpha^{(1)}$.
2: At $k^{th}$ iteration, given fixed $p_i^{(k-1)}$ and $\alpha^{(k-1)}$, estimate the following (in order)

    I. $m\big(X_i; \{p_i^{(k-1)}\}, \alpha^{(k-1)}\big)$ using (7).

    II. Solve $\quad \sum_{i=1}^{n} \frac{m(X_i; p_i^{(k-1)}, \alpha^{(k-1)})}{1 + \lambda \ m(X_i; \{p_i^{(k-1)}\}, \alpha^{(k-1)})} = 0$ for $\lambda$, which is a monotone function in $\lambda$.

    III. $p_i^{(k)} = \frac{1}{n} \frac{1}{1 + \lambda m(X_i; p_i^{(k-1)}, \alpha^{(k-1)})}, \forall i = 1, \ldots, n,$

    IV. $\alpha^{(k)} = \arg\max_\alpha \ \mathcal{L}_{Y,M,A|X}(\mathcal{D}; \alpha)$ subject to $w_a = 0$,
        where in $\mathcal{L}$, $\mathbb{E}[Y|X, A, M; \alpha_y] = w_0 + f(Z; \alpha_f) - \sum_{i=1}^{n} \big\{ \sum_m f(Z_i; \alpha_f) p(M|A = 0, X_i; \alpha_m) \big\} p_i^{(k)}$, and $f(Z) := \mathbb{E}[Y|X, A, M] - \mathbb{E}[Y|A, X = M = 0]$

3: Repeat Step (2) until convergence.

---

The resulting functional is then used as a moment restriction in the missing data version of the profile empirical likelihood in (6), yielding: (where $m_{\mathrm{mar}}(X; \alpha)$ is given in (7))

$$\underset{p_i, \alpha}{\arg\max} \prod_{i=1}^{n} p_i \times p(Y_i \mid M_i, A_i, X_i; \alpha_y)^{R_i} \times p(M_i \mid A_i, X_i; \alpha_m) \times p(A_i \mid X_i; \alpha_a) \times p(R_i \mid X_i, M_i, A_i; \alpha_r)$$
$$\text{such that} \quad \sum_{i=1}^{n} p_i = 1, \quad \sum_{i=1}^{n} p_i \times m_{\mathrm{mar}}(X_i; \alpha) = 0. \tag{8}$$

### 4.3. Fairness Constraints Via Hybrid Reparameterized Likelihoods

In Section 4.1, we reformulated the constrained optimization problem of interest by rewriting the likelihood in terms of the parameters we were interested in constraining, and directly setting those parameters to zero. However, we did not place any constraints on $p(X)$. In Section 4.2, we used hybrid likelihoods to constrain a nonparametric estimate of $p(X)$, but did not provide a convenient reparameterization of the likelihood in terms of relevant parameters. In this section we describe an approach to optimizing a *hybrid reparameterized likelihood* that combines the advantages of both proposals. This allows us to constrain the entire likelihood and do so with standard maximum likelihood software, since the constraint we must satisfy directly corresponds to a parameter in the hybrid likelihood.

For simplicity of presentation, we again focus on constraining the NDE in the full data version of the problem, although the methods we describe generalize without difficulty to constraints represented by arbitrary PSEs, and to the missing at random likelihood we use. The direct effect can then be estimated by $\mathbb{E}_x[m(X; \alpha)]$, where $m(X; \alpha)$ is given in (5), and $\mathbb{E}[Y|A, M, X; \alpha_y]$ is $\mathbb{E}[Y|Z; \alpha_y] = f(Z; \alpha_f) - \sum_{X,M} \{f(Z; \alpha_f) \times p(M|X, A = 0; \alpha_m) \times p(X)\} + w_0$. Once again, note that under MAR $\mathbb{E}[Y|Z; \alpha_y, R = 0] = \mathbb{E}[Y|Z; \alpha_y, R = 1]$. For an arbitrary PSE, $m(X; \alpha)$ is obtained from edge g-formula (Shpitser, 2013), and the outcome regression is reparameterized according to Theorem 2.

Assuming $p_i = p(X_i = x_i)$ as before, $m(X; \alpha)$ will be a function of $p_i$ parameters as well and we can use (7) to compute it. The profile empirical likelihood ratio ($\{p_i, \widehat{\alpha}\}^{opt}$) is then given by:

$$\underset{p_i, \alpha}{\arg\max} \prod_{i=1}^{n} p_i \times p(Y_i \mid M_i, A_i, X_i; \alpha_y)^{R_i} \times p(M_i \mid A_i, X_i; \alpha_m) \times p(A_i \mid X_i; \alpha_a) \times p(R_i \mid X_i, M_i, A_i; \alpha_r)$$

$$\text{such that} \quad \sum_{i=1}^{n} p_i = 1, \quad \sum_{i=1}^{n} p_i \times m_{\text{mar}}(X_i; p_i, \alpha) = 0. \tag{9}$$

Unlike the constrained optimization problem in (6), it is not straightforward to find the dual form of the optimization problem in (9), which is the standard approach for solving such problems in the empirical likelihood literature. The reason is that $p_i$ appears in multiple places – specifically, $m(X_i; p_i, \alpha)$ is now a function of both $\alpha$ and $p_i$. To solve this problem, we provide a heuristic approach for optimizing (9) via an iterative procedure that starts with an initialization of $\alpha$ and $p_i$s, and at the $k$th iteration updates the values for $\alpha^k$ and $p_i^k$s by treating $m(X_i; p_i, \alpha)$ as a function of $\{X_i, p_i^{k-1}, \alpha^{k-1}\}$. The procedure terminates when the difference between the two updates is sufficiently small. In Algorithm 1, we describe our proposed iterative procedure.

## 5. Experiments

**Simulation 1.** The result in Theorem 1 implies that the accuracy of our prediction procedure depends on which components of $p(Z, Y; \alpha)$ are constrained, which in turn is contingent on the chosen estimator $\widehat{g}(\mathcal{D})$. Here, we illustrate this dependence via experiments by considering four consistent estimators of the NDE presented in Tchetgen Tchetgen and Shpitser (2012). We generated synthetic data ($n = 5000$ with 20% missing outcomes), according to the causal model shown in Fig. 1(a), where $A, M$ are binary and $X, Y$ are continuous variables. We fit models for $\mathbb{E}[Y|A, M, X; \alpha_y]$, $p(M|A, X; \alpha_m)$, and $p(A|X; \alpha_a)$ by maximum likelihood. The first estimator (g-formula) is the MLE plug-in estimator and uses $Y$ and $M$ models to estimate NDE. The second one is the inverse probability weighted (IPW) estimator that uses $A$ and $M$ models. The third "mixed" estimator uses the $A$ and $Y$ models, and the fourth estimator (EIF) uses all three models, and is based on the efficient influence function for the PSE parameter. See Appendices C and D for details on these estimators and the model specifications. The code is attached to this submission.

We approximated the fair world $p^*$ by standard constrained MLE described in Section 2. We estimated the NDE using each of the four estimators and evaluated the performance of the approximated $p^*$ in each case. In Table 1, we show the estimated NDE with respect to $p^*$, the log likelihood, KL-divergence between $p^*$ and $p$, and the mean squared error (MSE) between the observed outcomes and the predicted ones (averaged over 100 repetitions). We contrast these results with the unconstrained prediction model. KL-divergence and MLE are reported with respect to $p(Y, M, A \mid X)$ since we

Table 1: *(left)* Comparing $p^*$ obtained via constraining different parts of the likelihood. *(right)* Evaluating different estimation methods by KL divergence and predictive accuracy (MSE). $\mathbf{M_0}$: Unconstrained MLE, $\mathbf{M_1}$: Constrained MLE (sec. 2.2), $\mathbf{M_2}$: Reparameterized MLE (sec. 4.1), $\mathbf{M_3}$: Hybrid MLE (sec. 4.2), and $\mathbf{M_4}$: Hybrid reparameterized MLE (sec. 4.3)

| Method | Estimator | Effect | Log $\mathcal{L}$ | $\mathbf{D_{KL}(p\|\|p^*)}$ | MSE |
|---|---|---|---|---|---|
| Unconstrained | g-formula | 2.19 | $-13148$ | 0.000 | 1.002 |
| Constrained | g-formula | 0.05 | $-15124$ | 0.395 | 3.459 |
| | IPW | 0.05 | $-13651$ | 0.101 | 5.009 |
| | Mixed | 0.05 | $-14348$ | 0.240 | 2.795 |
| | EIF | 0.05 | $-13560$ | 0.082 | 4.867 |

| Method | Effect | $\mathbf{D_{KL}(p\|\|p^*)}$ | MSE |
|---|---|---|---|
| $\mathbf{M_0}$ | 2.19 | 6.997 | 0.999 |
| $\mathbf{M_1}$ | 0.05 | 7.321 | 3.497 |
| $\mathbf{M_2}$ | 0.00 | 7.220 | 3.377 |
| $\mathbf{M_3}$ | 0.02 | 7.181 | 1.166 |
| $\mathbf{M_4}$ | 0.00 | 7.225 | 1.569 |

used an empirical evaluation of $p(X)$ in all the estimators of NDE. The role of constraining $p(X)$, via our described procedures in Section 4, is demonstrated in the next experiment. According to Table 1, unconstrained MLE is KL-closest to the true distribution and yields the lowest MSE, as expected. However, it suffers from being unfair: NDE $= 2.19$. In all the constrained MLE methods, NDE is restricted to lie between $-0.05$ and $0.05$. AIPW produces the second closest approximation to the true distribution while being fair, which is expected by Theorem 1. However, the MSE under AIPW is relatively large, since more information is averaged out from the predictions in $p^*$. The approximated fair distributions under the other three estimators are KL-farther from the true distribution, and the accuracy of prediction varies, underscoring how the performance of the learned model depends strongly on what part of the information is being averaged out and what estimator is being used.

**Simulation 2.** Here, we illustrate that even in simple settings our three proposed methods for solving constrained maximum likelihood problems considerably outperform the existing method described in Nabi and Shpitser (2018). We use the same synthetic data generated in Simulation 1, and assume that the direct effect of sensitive feature $A$ on outcome $Y$ is unfair. We estimate the effect via g-formula. We approximate the fair world $p^*$ by constrained MLE using the three methods described in Section 4, and contrast them with the constrained MLE described in Section 2 as well as regular unconstrained MLE. We evaluated the performance of all five methods by computing the direct effect with respect to $p^*$, KL-divergence between $p^*$ and $p$, and the MSE between the observed and predicted outcomes. Results are displayed in Table 1 (averaged over 100 repetitions). The NDE is again restricted to lie between $-0.05$ and $0.05$.

According to Table 1, our three proposed methods $(M_2, M_3, M_4)$, all yield a better approximation of the fair distribution $p^*$ compared to the standard constrained MLE $(M_1)$, in terms of KL-distance to the true unfair distribution $p$. Note that each likelihood method handles $p(X)$ differently: $M_0, M_1$, and $M_2$ do not constrain $p(X)$, while $M_3$ and $M_4$ directly include it in the constrained optimization – this explains the large KL difference between $p$ and $p^*$ (even for $M_0$) where here we are evaluating the distance to the entire joint $p(Y, M, A, X)$. The reparameterized MLE method in $M_2$ requires averaging over the constrained covariates. Hence, there is only minimal improvement in prediction accuracy (measured by MSE). However, both hybrid methods $M_3$ and $M_4$ use all information in the data, and therefore achieve substantial improvements in prediction accuracy.

**Simulation 3.** We emphasize the importance of Theorem 2 by generalizing the notion of direct effect as a measure of unfairness to a more complex path-specific effect involving multiple mediators. We generated data according to the model shown in Fig. 1(b) and assumed both the direct path and

the paths through $M$ are impermissible pathways. The reparameterized outcome regression, where the impermissible PSE shows up as a single parameter, and the corresponding optimization problem are shown in (3) and (4). The unfair PSE is estimated to be 2.39 and we restrict it to lie between $-0.05$ and $0.05$. The constrained MLE procedure in Nabi and Shpitser (2018) yields an MSE of 2.484, while our reparameterized MLE yields an MSE of 1.910 (averaged over 100 repetitions.) We observe further improvement in MSE by incorporating $p(X)$ into the constrained optimization problem, as suggested in Theorem 1, using our hybrid MLE procedure. As a result, the MSE reduces down to 1.131, highlighting the advantage of hybrid MLE procedure over regular constrained MLE.

## 6. Conclusion

Imposing hard fairness constraints on predictive models involves a balance of parametric modeling, nonparametric methods, and constrained optimization. In this paper, we have proposed two innovations to make the problem easier and make predictions more accurate: a reparameterization of the likelihood such that nonlinear constraints appear explictly as likelihood parameters constrained to be zero, and an incorporation of techniques from empirical likelihood theory to make the constrained distribution closer to the unconstrained unfair distribution. In addition to the immediate application in this paper, the reparameterization technique outlined in Theorem 2 solves a general open problem in causal mediation analysis. Though we focus primarily on the path-specific fairness constraints, the ideas presented here should be applicable more broadly to fair prediction proposals that require imposing constraints on predictive models. Our simulations show that even in a relatively simple setting, we can significantly improve on prior proposals, achieving prediction performance comparable to unconstrained (unfair) MLE, particularly with the hybrid approach. At this stage, our method which combines reparameterization with hybrid likelihood is somewhat heuristic; in future work, we hope to develop an approach for optimizing EL weights and likelihood parameters jointly without the need for iteration.

## Acknowledgments

We thank the anonymous reviewers for their comments. Ilya Shpitser is sponsored in part by NSF CAREER: 1942239, ONR N00014-21-1-2820, NIH R01 AI127271-01A1, and NSF 2040804.

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

## Supplementary Materials

In **Appendix A**, we provide two detailed examples to illustrate the reparameterization idea put forth in Theorem 2. In **Appendix B**, we provide a brief overview of empirical likelihood methods and some additional theoretical details useful for understanding our proposed hybrid likelihood approach. In **Appendix C**, we state the statistical modeling assumptions we made in our simulation experiments. In **Appendix D**, we give some relevant details for the simulations reported in the main paper. **Appendix E** contains proofs of our theorems.

## A. Reparameterized Likelihood: Examples

For reference, we reproduce the edge g-formula here from Shpitser and Tchetgen Tchetgen (2016):

$$p(V(\pi, a, a')) = \prod_{V_i \in V \setminus A} p(V_i \mid a \cap \mathrm{pa}_i^\pi, a' \cap \mathrm{pa}^{\overline{\pi}}, \mathrm{pa}_{\mathcal{G}}(V_i) \setminus A).$$

**Example 1.** Consider the DAG in Fig. 1(a), and assume the natural direct effect (NDE) is the unfair effect we wish to constrain to be $0$. The NDE corresponds to the counterfactual contrast of the form $Y(a, M(a'))$. Under certain identification assumptions discussed in Pearl (2001), the NDE is identified as follows.

$$
\begin{aligned}
\text{NDE} :=& E[Y(1, M(0))] - E[Y(0, M(0))] \\
=& \sum_{X,M} \mathbb{E}[Y \mid X, A = 1, M] \times p(M \mid A = 0, X) \times p(X) \\
& - \sum_{X,M} \mathbb{E}[Y \mid X, A = 0, M] \times p(M \mid A = 0, X) \times p(X)
\end{aligned}
\tag{10}
$$

According to Theorem 2, we get the following reparameterization of the regression model as follows.

$$
\begin{aligned}
\mathbb{E}[Y \mid X, A, M] =& \underbrace{\mathbb{E}[Y \mid X, A, M] - \mathbb{E}[Y \mid A, X = 0, M = 0]}_{f(X,A,M)} \\
& - \sum_{X,M} f(X, A, M) \times p(M \mid A = 0, X) \times p(X) \\
& + \underbrace{\sum_{X,M} \mathbb{E}[Y \mid X, A, M] \times p(M \mid A = 0, X) \times p(X)}_{\phi(A) = w_0 + w_a A}.
\end{aligned}
\tag{11}
$$

Note that the last term is only a function of $A$, and since $A$ is binary we can write it as $\phi(A) = w_0 + w_a A$. The coefficient $w_a$ corresponds to the direct effect, since

$$\text{NDE} = \sum_{X,M} \mathbb{E}[Y \mid X, A = 1, M] \times p(M \mid A = 0, X) \times p(X)$$

$$- \sum_{X,M} \mathbb{E}[Y \mid X, A = 0, M] \times p(M \mid A = 0, X) \times p(X)$$

$$= \phi(A = 1) - \phi(A = 0)$$

$$= w_a. \tag{12}$$

The observed data likelihood is given by

$$\mathcal{L}_{Y,M,A,X}(\mathcal{D}; \alpha)$$
$$= \prod_{i=1}^{n} p(Y_i | M_i, A_i, X_i; \alpha_y) \times p(M_i | A_i, X_i; \alpha_m) \times p(A_i | X_i; \alpha_a) \times p(X_i),$$

where $p(Y | M, A, X; \alpha_y)$ has mean given by eq. (11). The constrained optimization problem in eq. (2) then simplifies to the following optimization problem:

$$\arg \max_{\alpha} \ \mathcal{L}_{Y,M,A,X}(\mathcal{D}; \alpha) \quad \text{subject to} \quad w_a = 0. \tag{13}$$

In other words, we can simply set $w_a$ to be zero in the reparameterized outcome mean regression in eq. (11). Simply, $p(Y | M, A, X; \alpha_y)$ now has mean

$$\mathbb{E}[Y | X, A, M; \alpha_y] \tag{14}$$
$$= f(X, A, M; \alpha_f) - \sum_{x,m} f(X, A, M; \alpha_f) \times p(M | A = 0, X; \alpha_m) \times p(X) + w_0.$$

**Example 2.** Consider the DAG in Fig. 1(b), and assume the effect along the paths in $\{A \rightarrow Y, A \rightarrow M \rightarrow \cdots \rightarrow Y\}$ is the unfair path-specific effect (PSE) we wish to constrain to be 0. This PSE corresponds to the counterfactual contrast of the form $Y(a, M(a), L(a', M(a)))$. Under *no recanting witness* assumption Shpitser (2013), the PSE is identified as follows.

$$\text{PSE} := \mathbb{E}[Y(1, M(1), L(0, M(1)))] - \mathbb{E}[Y(0, M(0), L(0, M(0)))] \tag{15}$$
$$= \sum_{X,M,L} \mathbb{E}[Y \mid X, A = 1, M, L] \times p(L \mid A = 0, X, M) \times p(M \mid A = 1, X) \times p(X)$$
$$- \sum_{X,M,L} \mathbb{E}[Y \mid X, A = 0, M, L] \times p(L \mid A = 0, X, M) \times p(M \mid A = 0, X) \times p(X).$$

According to Theorem 2, we get the following reparameterization of the regression function.

$$\mathbb{E}[Y \mid X, A, M, L] = \underbrace{\mathbb{E}[Y \mid X, A, M, L] - \mathbb{E}[Y \mid A, X = M = L = 0]}_{f(X,A,M,L)} \qquad (16)$$

$$- \sum_{X,M,L} f(X, A, M, L) \times p(L \mid A = 0, X, M) \times p(M \mid A, X) \times p(X)$$

$$+ \underbrace{\sum_{X,M,L} \mathbb{E}[Y \mid X, A, M, L] \times p(L \mid A = 0, X, M) \times p(M \mid A, X) \times p(X)}_{\phi(A)=w_0+w_a A}.$$

Similar to Example 1, the last term in the display above, is only a function of $A$ and can be written as $w_0 + w_a A$, if $A$ is binary. Given the identification functional in eq. (15), it is straightforward to show that the coefficient $w_a$ corresponds to the path-specific effect that we want, i.e.,

$$\text{PSE} = \phi(A = 1) - \phi(A = 0) = w_a.$$

The observed data likelihood is given by

$$\mathcal{L}_{Y,L,M,A,X}(\mathcal{D}; \alpha) = \prod_{i=1}^{n} p(Y_i|L_i, M_i, A_i, X_i; \alpha_y) \times p(L_i \mid M_i, A_i, X_i; \alpha_l)$$

$$\times p(M_i|A_i, X_i; \alpha_m) \times p(A_i|X_i; \alpha_a) \times p(X_i),$$

where $p(Y|L, M, A, X; \alpha_y)$ has mean given by eq. (16). The constrained optimization problem in eq. (2) then simplifies to the following optimization problem:

$$\arg\max_{\alpha} \ \mathcal{L}_{Y,L,M,A,X}(\mathcal{D}; \alpha) \quad \text{subject to} \quad w_a = 0.$$

In other words, we can simply set $w_a$ to be zero in the reparameterized outcome mean regression in eq. (16). Simply, $p(Y|L, M, A, X; \alpha_y)$ now has mean

$$\mathbb{E}[Y|X, A, M, L; \alpha_y]$$
$$= f(X, A, M, L; \alpha_f)$$
$$- \sum_{x,m,l} f(X, A, M, L; \alpha_f) \times p(L|A = 0, M, X; \alpha_l) \times p(M|A, X; \alpha_m) \times p(X)$$
$$+ w_0.$$

## B. Hybrid Likelihood: Overview and Details

### Empirical Likelihood

We briefly review empirical likelihood methods, described in detail in Owen (2001). Let $X_1, \ldots, X_n$ be independent random vectors with common distribution $F_0$. Let $F$ be any CDF, where $F(x) =$

$p(X \leq x)$, and $F_n$ be the empirical distribution. Suppose that we are interested in $F$ through $\theta = T(F)$, where $T$ is a real-valued function of the distribution. The true unknown parameter is $\theta_0 = T(F_0)$. Proceeding by analogy to parametric MLE, the non-parametric MLE of $\theta$ is $\hat{\theta} = T(F_n)$. The nonparametric likelihood ratio, $R(F) = \frac{\mathcal{L}(F)}{\mathcal{L}(F_n)}$, is used as a basis for hypothesis testing and deriving confidence intervals. The *profile likelihood ratio* function is defined as

$$\mathcal{R}(\theta) = \sup \big\{ R(F) \mid T(F) = \theta, F \in \mathcal{F} \big\},$$

where $\mathcal{F}$ denotes the set of all distributions on $\mathbb{R}$.

Often, $\theta \equiv \theta(F)$ is the solution to an estimating equation of the form $\mathbb{E}[m(X, \theta)] = 0$. A natural estimator for $\theta$ is produced by solving the empirical estimating equation $\frac{1}{n} \sum_{i=1}^{n} m(X_i, \hat{\theta}) = 0$. Assuming $p_i = f(X = x_i)$ for $i = 1, \ldots, n$, the *profile empirical likelihood ratio* function of $\theta$ is defined as

$$\mathcal{R}(\theta) = \max \Big\{ \prod_{i=1}^{n} p_i \quad \text{such that} \quad \sum_{i=1}^{n} p_i \times m(X_i, \theta) = 0, \quad p_i \geq 0, \quad \sum_{i=1}^{n} p_i = 1 \Big\}. \quad (17)$$

Since maximizing the likelihood is equivalent to maximizing the logarithm of the likelihood, the profile empirical likelihood ratio is rewritten in terms of log likelihood as follows.

$$\mathcal{R}(\theta) = \max \Big\{ \sum_{i=1}^{n} \log p_i \quad \text{such that} \quad \sum_{i=1}^{n} p_i \times m(X_i, \theta) = 0, \quad p_i \geq 0, \quad \sum_{i=1}^{n} p_i = 1 \Big\}. \quad (18)$$

In order to solve the above optimization problem, we can apply the Lagrange multiplier method.

$$T(\{p_i\}, \lambda, \lambda_1) = \sum_{i=1}^{n} \log p_i + \lambda_1 (\sum_{i=1}^{n} p_i - 1) - n\lambda \sum_{i=1}^{n} p_i \times m(X_i; \theta),$$

where $\lambda, \lambda_1$ are the Lagrange multipliers. We take the derivative of $T(\{p_i\}, \lambda, \lambda_1)$, with respect to the $p_i$'s, and set them to zero. Solving the system of equations reveals that $\lambda_1 = -n$, and

$$p_i = \frac{1}{n} \times \frac{1}{1 + \lambda m(X_i; \theta)}, \quad \forall i = 1, \ldots, n, \quad (19)$$

where $\lambda$ is the solution to

$$\sum_{i=1}^{n} \frac{m(X_i; \theta)}{1 + \lambda \, m(X_i; \theta)} = 0, \quad (20)$$

which is a monotone function in $\lambda$. Maximizing the profile empirical log-likelihood ration in (18) is equivalent to maximizing the following (substituting $p_i$ from (19) into (18)):

$$l(\theta) = - \sum_{i=1}^{n} \log(1 + \lambda \, m(X_i; \theta)) - n \log n. \quad (21)$$

Maximizing $l(\theta)$ over a small set of parameters $\theta$, is a much simpler optimization problem than maximizing (18) over $n$ unknowns. Equation 21 is known as the dual representation of 18. See Owen (2001) for more details.

**Hybrid Likelihood**

Now, consider independent pairs $(X_1, Y_1), \ldots, (X_n, Y_n)$. Suppose that all $n$ observations are independent, and that we have a correctly specified parametric model for $p(Y|X; \theta_y)$ but $p(X)$ is unspecified. Let $p_i = p(X = x_i)$. A natural approach for estimating $\theta_y$ and the $p_i$s is to form a *hybrid* likelihood that is nonparametric in the distribution of $X_i$ but is parametric in the conditional distribution of $Y_i|X_i$:

$$\mathcal{L}(\mathcal{D}; \{p_i\}, \theta) = \prod_{i=1}^{n} p_i \times p(Y_i|X_i; \theta).$$

Suppose we are interested in parameter $\theta$ through the estimating equation $\mathbb{E}[m(X, Y; \theta)] = 0$. Hence, the equivalent form of (18) for the profile hybrid likelihood ratio function is as follows:

$$\mathcal{R}(\theta) = \max \left\{ \sum_{i=1}^{n} \left( \log p_i + \log p(Y_i|X_i; \theta) \right) \right.$$

$$\text{such that} \quad \sum_{i=1}^{n} p_i \times m(X_i, Y_i; \theta) = 0, \quad p_i \geq 0, \quad \left. \sum_{i=1}^{n} p_i = 1 \right\}. \tag{22}$$

Similar to the empirical likelihood, we can apply the Lagrange multiplier method to solve the above optimization problem. For more details, see Owen (2001) and Qin (2017).

## C. Simulation details

Here we report the precise parameter settings used in our simulation studies. We trained our models on a batch size of $5,000$ using the following data generating processes, where outcome $Y$ is treated as missing on $20\%$ of the data.

**Simulations 1 and 2.** In these simulations, data is generated according to the causal model shown in Fig. 1(a) as follows.

$$X \sim \mathcal{N}(0, 1)$$
$$p(A = 1 \mid X) \sim \text{expit}(-0.5 - 0.5X)$$
$$p(M = 1 \mid A, X) \sim \text{expit}(-0.5 - X - 0.5A + AX)$$
$$Y = 1 + X + 2A - 2AX + M + 3XM + AM + XAM + \mathcal{N}(0, 1)$$

**Simulation 3.** In this simulation, data is generated according to the causal model shown in Fig. 1(b) as follows. $X, A,$ and $M$ are generated in the same way as the ones above.

$$p(L = 1|A, X, M) \sim \text{expit}(-0.5 - X - 0.5A - 0.25M + AX + 0.5AM + 0.25AXM)$$
$$Y = 1 + X + 2A + M + 0.5L - 2AX + AM + AL + AML + \mathcal{N}(0, 1)$$

## D. Details on Estimation Strategies

Given Theorem 1, the accuracy of the prediction procedure will depend on what parts of $p(Z, Y; \alpha)$ are constrained, and following Nabi and Shpitser (2018) this depends on the estimator $\widehat{g}(\mathcal{D})$. Here, we define several consistent estimators of the NDE (assuming the model shown in Fig. 1(a) is correct) presented in Tchetgen Tchetgen and Shpitser (2012).

**G-formula**: The first estimator is the MLE plug in estimator, where we use the $Y$ and $M$ models to estimate NDE. We fit models $\mathbb{E}[Y|A, M, X; \alpha_y]$ and $p(M|A, X; \alpha_m)$ by maximum likelihood, and use the following formula:

$$\mathbb{P}_n\bigg( \sum_m \Big( \mathbb{E}[Y_i \mid A = 1, X_i, M; \widehat{\alpha}_y] - \mathbb{E}[Y_i \mid A = 0, X_i, M; \widehat{\alpha}_y] \Big) \times p(M \mid A = 0, X_i; \widehat{\alpha}_m) \bigg).$$
(23)

Since solving (2) using (23) entails constraining $\mathbb{E}[Y|A, M, X]$ and $p(M|A, X)$, classifying a new instance entails using $\mathbb{E}[Y|A, X] = \sum_M \mathbb{E}[Y|A, M, X] \times p(M|A, X)$.

**Inverse probability weighting (IPW)**: The second estimator is the IPW estimator where we use the $A$ and $M$ models to estimate NDE. We can fit the models $p(A|X; \alpha_a)$ and $p(M|A, X; \alpha_m)$ by MLE, and use the following weighted empirical average as our estimate of the NDE:

$$\mathbb{P}_n\left( \frac{\mathbb{I}(A_i = 1)}{p(A_i = 1|X_i; \widehat{\alpha}_a)} \times \frac{p(X_i|A = 0, X_i; \widehat{\alpha}_m)}{p(M_i|A = 1, X_i; \widehat{\alpha}_m)} \times Y_i - \frac{\mathbb{I}(A_i = 0)}{p(A_i = 0|X_i; \widehat{\alpha}_a)} \times Y_i \right).$$
(24)

Since solving the constrained MLE problem using this estimator entails only restricting parameters of $A$ and $M$ models, predicting a new instance is done using $\mathbb{E}[Y|X] = \sum_{A,M} \mathbb{E}[Y|A, M, X] \times p(M|A, X) \times p(A|X)$.

**Mixed approach**: The third way of computing the NDE is using $A$ and $Y$ models. In this estimator, we fit the models $p(A|X; \alpha_a)$ and $\mathbb{E}[Y|A, M, X; \alpha_y]$ by MLE, as usual, and combine the edge G-formula and IPW in the following way:

$$\mathbb{P}_n\left( \frac{\mathbb{I}(A_i = 0)}{p(A_i = 0|X_i; \widehat{\alpha}_a)} \times \mathbb{E}[Y_i|A = 1, M_i, X_i; \widehat{\alpha}_y] - \mathbb{E}[Y_i|A = 0, M_i; \widehat{\alpha}_y] \right),$$
(25)

Since solving the constrained MLE problem using this estimator entails only restricting parameters of $A$ and $Y$ models, predicting a new instance is done using $\mathbb{E}[Y|M, X] = \sum_A \mathbb{E}[Y|A, M, X] \times \frac{p(M|A,X) \times p(A|X)}{\sum_A p(M|A,X) \times p(A|X)}$.

**Efficient Influence Functinon (EIF)**: The final estimator uses all three models, as follows:

$$\mathbb{P}_n\bigg(\frac{\mathbb{I}(A_i = 1)}{p(A_i = 1|X_i; \widehat{\alpha}_a)} \times \frac{p(M_i \mid A = 0, X_i; \widehat{\alpha}_m)}{p(M_i|A = 1, X_i; \widehat{\alpha}_m)} \times \Big\{Y_i - \mathbb{E}[Y_i|A = 1, M_i, X_i; \widehat{\alpha}_y]\Big\} \quad (26)$$

$$+ \frac{\mathbb{I}(A_i = 0)}{p(A_i = 0|X_i)} \times \Big\{\mathbb{E}[Y_i|A = 1, M_i, X_i; \widehat{\alpha}_y] - \eta(1, 0, X_i)\Big\} + \eta(1, 0, X_i)$$

$$- \frac{\mathbb{I}(A_i = 0)}{p(A_i = 0|X_i; \widehat{\alpha}_a)} \times \Big\{Y_i - \eta(0, 0, X_i)\Big\} + \eta(0, 0, X_i)\bigg),$$

with $\eta(a, a', X) \equiv \sum_M \mathbb{E}[Y|a, M, X] \times p(M|a', X)$. Since the models of $A$, $M$, and $Y$ are all constrained with this estimator, predicting $Y$ for a new instance is via $\mathbb{E}[Y|X] = \sum_{A,M} \mathbb{E}[Y|A, M, X] \times p(M|A, X) \times p(A|X)$. For more details on semiparametric estimators of average causal effects in presence of unmeasured confounders, see Bhattacharya et al. (2020).

## E. Proofs

**Theorem 1** *Let $p(Z)$ denote the observed data distribution and let $Z_1, Z_2 \subseteq Z$. Let $p_1^*(Z)$ and $p_2^*(Z)$ denote two constrained distributions that are obtained by constraining the distribution over variables that are not in $Z_1$ and $Z_2$, respectively. That is $p_1^*$ constrains the variables in $Z \setminus Z_1$ and $p_2^*$ constrains the variables in $Z \setminus Z_2$, so $p_1^*(Z_1) = p(Z_1)$ and $p_2^*(Z_2) = p(Z_2)$. More formally,*

$$p_1^*(Z) = \arg\min_{q(Z)} D_{KL}(p \mid\mid q), \text{ s.t. } \epsilon_l \leq g(q(Z)) \leq \epsilon_u, \text{ and } q(Z_1) = p(Z_1),$$

$$p_2^*(Z) = \arg\min_{q(Z)} D_{KL}(p \mid\mid q), \text{ s.t. } \epsilon_l \leq g(q(Z)) \leq \epsilon_u, \text{ and } q(Z_2) = p(Z_2).$$

*If $Z_2 \subseteq Z_1 \subseteq Z$, then $D_{KL}(p \mid\mid p_2^*) \leq D_{KL}(p \mid\mid p_1^*)$.*

**Proof** Let

$$M_1 = \Big\{p_1^*(Z) = \arg\min_{q(Z)} D_{KL}(p||q), \text{ s.t. } \epsilon_l \leq g(q(Z)) \leq \epsilon_u, \text{ and } q(Z_1) = p(Z_1)\Big\},$$

and

$$M_2 = \Big\{p_2^*(Z) = \arg\min_{q(Z)} D_{KL}(p||q), \text{s.t. } \epsilon_l \leq g(q(Z)) \leq \epsilon_u, \text{ and } q(Z_2) = p(Z_2)\Big\}.$$

Since the joint distribution in model $M_1$ is more restricted than $M_2$, then $M_1$ is a submodel of $M_2$. This implies that maximizing the likelihood under model $M_1$ yields a likelihood that is less than or equal to the one under model $M_2$, i.e., $\max \mathcal{L}_{M_1}(\mathcal{D}) \leq \max \mathcal{L}_{M_2}(\mathcal{D})$. Maximizing the likelihood of observed data with respect to the model parameters is equivalent to minimizing KL-divergence between the likelihood and the true distribution of the data (Wasserman, 2013). Consequently, KL-divergence between $p^*$ and $p$ is smaller in $M_2$ compared to $M_1$, i.e $D_{KL}(p \mid\mid p_2^*) \leq D_{KL}(p \mid\mid p_1^*)$. ■

**Theorem 2** *Assume the observed data distribution $p(Y, Z)$ is induced by a causal model where $Z = \{X, A, M\}$ includes pre-treatment measures $X$, binary treatment $A$, and post-treatment pre-outcome mediators $M$. Let $p(Y(\pi, a, a'))$ denote the potential outcome distribution that corresponds*

*to the effect of $A$ on $Y$ along proper causal paths in $\pi$, where $\pi$ includes the direct edge $A \to Y$, and let $p(Y_0(\pi, a, a'))$ denote the identifying functional for $p(Y(\pi, a, a'))$ obtained from the edge g-formula, where the term $p(Y|Z)$ is evaluated at $\{Z \setminus A\} = 0$. Then $\mathbb{E}[Y|Z]$ can be written as follows:*

$$\mathbb{E}[Y|Z] = f(Z) - \big(\mathbb{E}[Y(\pi, a, a')] - \mathbb{E}[Y_0(\pi, a, a')]\big) + \phi(A),$$

*where $f(Z) := \mathbb{E}[Y|Z] - \mathbb{E}[Y|A, \{Z \setminus A\} = 0]$ and $\phi(A) = w_0 + w_a A$. Furthermore, $w_a$ corresponds to $\pi$-specific effect of $A$ on $Y$.*

**Proof** By letting $\phi(A = a) = \mathbb{E}[Y(\pi, a, a')]$, it suffices to show that $\mathbb{E}[Y_0(\pi, a, a')] = \mathbb{E}[Y|A, \{Z \setminus A\} = 0]$. Given the identification result for edge-consistent counterfactuals in Shpitser and Tchetgen Tchetgen (2016), we can write the identification functional as follows.

$$\mathbb{E}[Y_0(\pi, a, a')] = \sum_{V \in \mathfrak{X}_V \setminus \{A,Y\}} \mathbb{E}[Y|A = a, \{Z \setminus A\} = 0] \times h(V \in \mathfrak{X}_V \setminus Y),$$

where $h(V \in \mathfrak{X}_V \setminus Y)$ is a function of all variables excluding $Y$. Note that $h$, does not include any density where $A$ appears on the LHS of the conditioning bar. Therefore, we have:

$$\mathbb{E}[Y_0(\pi, a, a')] = \mathbb{E}[Y|A = a, \{Z \setminus A\} = 0] \times \sum_{V \in \mathfrak{X}_V \setminus \{A,Y\}} h(V \in \mathfrak{X}_V \setminus Y)$$

$$= \mathbb{E}[Y|A = a, \{Z \setminus A\} = 0].$$

∎

