# OpenReview forum: "Optimal Training of Fair Predictive Models"
_cclear.cc/CLeaR/2022/Conference — CLeaR 2022 Oral_

### Official Review · Reviewer_ujaL · 2021-11-21

**Confidence:** 4
**Overall Score:** 6

**Main Review:**

The topic of this work is significant. How to impose hard fairness constraints on predictive models and optimize them efficiently is a bottleneck in fairness learning. Below are some concerns:

Concerns:
1.	The problem this work want to address is not clearly defined. In the second paragraph of Section 3, the authors aim to predict the outcome labels for new instances, subject to fairness constraints. Are new instances the test data? Then, the authors train on the combination of historical data and new instances. Does this mean the test data is used in the training phase?
2.	The motivation of this work is confusing. What is the distribution $p^{*}(Y, Z)$ of the fair world? What is the distribution $p(Y, Z)$ of the unfair world? Z is the features. Why would Z change in the two worlds?
3.	What is the definition of $g(p(Y, Z))$? Why in practice, it should be strictly bounded as 0? What is the variables that appear in the estimator $\widehat{g}(\mathcal{D})$? Doesn’t $\mathcal{D}$ contain the whole Z (Z1+Z2)?
4.	In your Theorem, the observed data distribution $p(Y, Z)$ is induced by a causal model. Is there any requirements for the causal model? Has it to be in the form of Figure 1?
5.	This paper is not well organized, e.g., the Fig 1 is referred to in page 2 but located in page 5. The logic and connection among those sections is ambiguous.
6.	Some grammatical issues, e.g., `… pathways are … context-specific, hence require input ...’ and typos, e.g., $R_i$ in the formula at the bottom of page 5 is out of the bracket, should be fixed.
7.	How will this method perform on real-world dataset?


**Summary:**

In this work, based on Nabi and Shpitser (2018), the authors introduce new theoretical results and optimization techniques to make model training easier and more accurate. Experiments on simulations show the effectiveness of this method.

---

> ### Author Response · Authors · 2021-11-30
> **Reviewer ujaL**
>
> We thank the reviewer for their positive comments, and questions.
>
> Q1: Training and test data
>
> A: Our training and test data do not overlap.
>
> Q2: Differences between fair p*(Y, Z) and unfair p(Y, Z) worlds. Does Z change?
>
> A: The covariates Z do not change between the two worlds. However, their distribution may change depending on how we approximate the fair distribution p*(Y, Z). Intuitively, the idea is that in the counterfactual world where our fairness constraint is satisfied, there are associations or dependencies among variables that are different somehow, and this induces a possible different joint distribution over Y, Z.
>
>
> Q3: What is the definition of g(p(Y, Z))? Why in practice, it should be strictly bounded as 0? What is the variables that appear in the estimator g^(D)? Doesn’t D contain the whole Z (Z1+Z2)?
>
> A: Our measure of fairness is the effect along unfair pathways. A path-specific effect is a counterfactual quantity that first needs to be identified as a function of observed data. We denote the identified functional for our unfair effect generically via g(p(Y, Z)), where p(Y, Z) is the observed data distribution. Different specific choices of g correspond to different functionals. The functional g(p(Y, Z)) needs to be computed/estimated from data which we denote by g^(D). D here just highlights the fact that the g functional is computable using samples in D. In practice, bounds near zero (or “null” value) are desirable, though the tolerance bounds (\epsilon^-, \epsilon^+ )are left up to the user.
>
> Q4: Is there any requirement for the causal model?
>
> A: Our assumption is that the observed data distribution factorizes wrt a directed acyclic graph (where some variables may be unmeasured).
>
> Q5: Fig 1 is referred to in page 2 but located in page 5. The logic and connection among those sections is ambiguous.
>
> A: This is an artifact of latex compilation. We are happy to move the figure closer to where it’s first referenced.
>
> Q6: Grammatical issues and typos
>
> A: Thanks for catching these. We will fix them in the camera-ready.
>
> Q7: How will this method perform on real-world dataset?
>
> A: Given the outlined theory, our expectation is that the methods we proposed should lead to better predictive performance in fair predictive modeling. Our simulations confirm that even in simple scenarios, this is true. Though we do not have space to do so here, we hope to apply these methods to real data in a nearby future work.

---

### Official Review · Reviewer_sqKv · 2021-11-22

**Confidence:** 3
**Overall Score:** 8

**Main Review:**

In this paper, the authors proposed a new method to train the fair predictive models. The new approach is guaranteed to achieve more accurate estimate and is easier to optimize than the one from (Nabi and Shpitser 2018).

1. The paper is very well written and easy to understand. I did not check the proof in details, but as far as I am aware, the proofs are correct and technically sound;

2. The simulations are solid and very comprehensive. It would be nicer if the authors can provide a easier to use computational toolbox from their proposed method. This can create a broader impact of the proposed work;

3. The authors can consider citing the other results aiming for fairness in computer assisted decision making, such as the one from
https://arxiv.org/abs/2110.15310 and the references therein. In particular, it would also be interesting to compare the criteria of fairness used in the two papers.

Overall, as I am not an expert of fairness, and I did not keep a close track of the most recent publications in fairness, my comments might be biased. Still I believe this is a good paper!

**Summary:**

A very strong piece of work!

---

> ### Author Response · Authors · 2021-11-30
> **Reviewer sqKv**
>
> We thank the reviewer for their positive comments.
>
> Regarding the toolbox: We will make our code publicly available. We have thought about creating a fairness toolbox, but it certainly requires more time.
>
> Regarding the citation and discussion of the suggested paper: Thanks for sharing the reference. We are happy to add it to our references and discuss it in the paper.

---

### Official Review · Reviewer_upxr · 2021-11-24

**Confidence:** 2
**Overall Score:** 6

**Main Review:**

The paper studies an approach to simplify the computation of prediction functions under constraints on path-specific effects. For the case where the the direct effect of the treatment to the outcome variable is in the paths that must be controlled, they present a reparametrization of the likelihood function that makes explicit the total path-specific effect that should be controlled. The advantage of this reformulation is to allow for simpler algorithm for the optimization of the likelihood under fairness constraints. The authors then discuss an extension of their model to learn the full distribution (including P(x))

pros: the paper presents the problem and approach thoroughly. Theorem 2 seems non-trivial and rather strong.

cons: overall I found the paper hard to follow and I had difficulties understanding the conclusions of the experiments:
* As for many papers on controlling path-specific effects, the notation and the formulas are quite heavy, but I guess this is somewhat unavoidable. Also, and similarly to other papers on controlling path-specific effects, the paper discusses only a few very simple causal models. While the generality of the theoretical results is still there, it is unclear how practical that is.
* I did not understand the take-away message of Simulation 1 (Actually, I am not sure to understand what part of the paper is assessed in Simulation 1 and how it relates to simulation 2).
* The authors consider a transductive setting (we know the points to be labeled), and there is no reason for transductive learning to be better than inductive learning in general. It is unclear to me why the estimation of  P(X) turns out to improve predictive performances on this problem.
* In general, I found it unclear what was the aim of the authors -- improving predictive performance given a fairness constraint, or finding a better approximation to the true distribution that respects the fairness constraints. From the experimental results, there is no algorithm that performs better on the two criteria, and it is unclear to me whether this was expected or whether the experimental results are inconclusive.

**Summary:**

review

---

> ### Author Response · Authors · 2021-11-30
> **Reviewer upxr**
>
>
> We thank the reviewer for their positive comments, and questions.
>
> Q1: Notations and examples
>
> A: We understand expressing counterfactual path-specific effects can be notationally hard to digest. However, they have clear interpretations in terms of decomposing the total effect along various causal mechanisms. We provided a few examples, both in the main draft and in the appendix, to better illustrate our points and help the readers with the notations. This however, does not imply our proposed methods are only limited to the examples we provided. We are happy to discuss more complicated models in the appendix (or in the main draft as space permits).
>
>
> Q2: What are the take-away messages of simulation 1? (Actually, I am not sure to understand what part of the paper is assessed in Simulation 1 and how it relates to simulation 2)
>
> A: Simulation 1 is related to Theorem 1. There are various ways of estimating an identified causal functional (likelihood-based methods, inverse probability weighting approaches, influence function based methods, etc). Our choice of estimator for computing unfair effects will affect the performance of our constrained optimizer. This is because different estimators rely on different parts of the joint distribution. What we showed in Theorem 1 is that if a larger part of the joint is being constrained, the constrained distribution then lies closer (in the KL-divergence sense) to the unconstrained distribution. In Simulation 1, for illustration our unfair effect is the direct effect. Estimating the direct effect with the influence function based estimator relies on most of the joint distribution; and hence leads to an approximation of the fair distribution that is the closest to the unfair distribution, as opposed to other scenarios where we pick a different estimator.
> Simulation 2 is related to Theorem 2. This shows that if we fix our estimator (say we use g-formula) then additionally constraining p(X), on top of other parts that appear in g-formula, moves our approximation of the fair distribution closer to the unfair distribution.
>
> Q3: Why does estimation of P(X)  improve predictive performance?
>
> A: This is a consequence of what we show in Theorem 1. On one hand, the unconstrained distribution has the best predictive performance, but it does not enforce our fairness constraints. On the other hand, constraining more parts of the joint distribution moves us closer to the unconstrained distribution. Hence, by constraining P(X) we hope to move closer to the unconstrained distribution and achieve better predictive performance.
>
> Q4: Is the objective to improve predictive performance given a fairness constraint, or finding a better approximation to the true distribution that respects the fairness constraints?
>
> A: It’s a mix of the two. We want to achieve as best predictive performance as possible while satisfying our (hard) fairness constraints. Nearer approximations to the unconstrained distribution will achieve better predictive performance (since the unconstrained version will always do best).

---

### Decision · Program_Chairs · 2022-01-12

**Decision:**

Accept (Oral)

**Comment:**

The paper proposes an approach to facilitate the learning of predictive models satisfying certain (path specific) fairness constraints, by introducing a reparameterization of the likelihood to reduce the computational complexity of the associated optimisation problem, in combination with empirical likelihood techniques to improve the predictive performance. Key theorem 2 behind the method is also relevant to mediation analysis in general.

All reviewers were clearly in favour of acceptance, albeit with moderate confidence. The latter is likely primarily due to the technically and conceptually quite challenging ideas behind the notion of path-specific fairness and the resulting optimisation problem to learn the target fair prediction model. In that sense the main aim of the paper - reducing the complexity of tackling this problem in practice - is a strong argument for its relevance. Whether or not it is also sufficient to bring ‘path-specific fair prediction models’ to the masses is doubtful, as it still remains a hard problem, but it is definitely a step in the right direction. Nevertheless it is a pity that the apparent intuitive simplicity of ‘fairness’ does not lead to an equally transparent solution that would reassure non-experts as well.

Pros:
- tackles important problem in a highly topical subject
- good technical quality
- non-trivial results backed up by experimental support
Cons:
- clarity remains the main issue: despite attempts by the authors the paper will still be hard to access for most researchers not familiar with the subject of path-specific fairness (although the ideas should translate to other notions of fairness as well)
- experimental results are limited and hard to generalise

On the whole the benefits clearly outweigh the drawbacks, and therefore I recommend accept, with slightly lowered confidence to reflect the lingering uncertainty among the reviewers. Given the amount of general interest in fairness and related aspects I also recommend an oral presentation to give the authors an opportunity to present their work in a more easily accessible format.